# OpenReview forum: "Representation Drift Compensation: A Near-Zero Inference Cost Enhancement for LLM Decomposition"
_ICML.cc/2026/Conference — ICML 2026 regular_

### Official Review · Reviewer_pQu6 · 2026-03-02

**Soundness:** 3
**Presentation:** 2
**Significance:** 3
**Originality:** 2
**Overall Recommendation:** 4
**Confidence:** 4

**Summary:**

This paper explores the extension of low-rank decomposition for accelerating LLMs and MLLMs. It introduces the concept of "representation drift", demonstrating that applying low-rank decomposition within Transformer architectures disrupts block-level representations. To mitigate this drift, the authors propose the "Decomper" module. Essentially, this module utilizes calibration data to learn a bias compensation for the decomposed low-rank linear operators. This compensation is subsequently integrated back into the operators, achieving a solution with zero extra overhead during inference. Extensive evaluations across various backbones and tasks demonstrate the effectiveness of the proposed method.

**Compliance With Llm Reviewing Policy:**

Affirmed.

**Final Justification:**

The paper addresses a practically important problem in low-rank decomposition for LLMs named representation drift. Its empirical study is one of its main strengths, the experiments are generally well executed. The proposed compensation mechanism is easy to implement and appears effective in practice, which makes this work potentially useful for real deployment.

My main concern in initial review, however, remains the depth of the theoretical contribution. In particular, Section 3 does not yet provide sufficiently strong justification for why a learned bias-style compensation should be the right and correct the proposed representation drift. While the rebuttal clarify several points, it does not fully resolve my central concern about the necessity of this core design choice. The current analysis still feels more like an intuitive motivation than a convincing theory. Moreover, I believe the paper would benefit from a clearer positioning against closely related correction/update strategies (e.g. LoRA-like parameterization which is mentioned by other reviewer, or direct update A/B).

The rebuttal is helpful (including additional experiments for other reviewers) and does change my evaluation. The authors provide basically reasonable clarifications, acknowledge some phrasing issues, and add extra empirical evidence. Although I remain unconvinced about the theoretical part (**to be honest, the theoretical analysis in Section 3 appears preliminary, and the authors further responses are kind of evasive in my view**), I do believe the paper offers a practical contribution: it identifies a meaningful failure mode in LLM decomposition, proposes a lightweight compensation mechanism, and demonstrates empirical gains at low training cost.

Overall, the rebuttal partially addresses my concerns and improved my confidence in the practical value of the method. For this reason, I raise my score to **4 (Weak Accept)**, mainly in **recognition of the paper’s practical contribution and empirical effectiveness**, while still encouraging the authors to substantially strengthen the analysis in Section 3 in the revised version.

**Key Questions For Authors:**

Please refer to weaknesses.

**Limitations:**

yes

**Strengths And Weaknesses:**

**Strengths**
1. The experimental evaluation in this paper is comprehensive, demonstrating the authors' deep familiarity with this field. The experimental design incorporates most of the analytical evaluations typically required in literature regarding efficient LLM inference.
2. The proposed method is straightforward to implement. Despite some deficiencies in the paper's exposition, if the reported results hold true, this approach holds significant value for industrial applications aiming to accelerate model inference via decomposition.
3. The authors evaluated the method across a broad spectrum of model families, including Llama2/3, Qwen3, OPT, and Qwen2.5-VL. The results consistently demonstrate the effectiveness of the proposed approach.

**Weaknesses**
1. Model naming consistency is required throughout the paper: "QWen" should be standardized to "Qwen".
2. Although the paper attempts to define a phenomenon called "representation drift" and illustrates some of its characteristics experimentally, the description of this concept bears a strong resemblance to the theory presented in OBC [1]. I suggest summarizing the overlapping derivations and concepts into a "Preliminary" section. This would help clearly distinguish the key innovations of the representation drift theory. Specifically, what are the respective distributions of the propagated error and the local error? Can the propagated error be ignored? Exactly which type of error is $c$ compensating for?
3. The statement "we find its loss landscape to be well-behaved in practice, enabling the effective optimization of the compensation vector $c$" lacks sufficient experimental and theoretical support. Given that this is the fundamental motivation for the compensation vector $c$ scheme, rigorous proof and analysis are required.
4. "Zero-Cost deployment" is a merit of the proposed method. However, regarding the formula $b^′=b+c$, a question arises: What if we directly optimize the bias parameter (using $b$ as the initialization) via back-propagation on the calibration dataset? If this baseline approach also yields comparable results, the necessity of introducing the additional vector $c$ would be significantly diminished.
5. Why other potential compensation options (e.g. compensation for A and B) are not chosen?

References
- [1] Optimal Brain Compression: A Framework for Accurate Post-Training Quantization and Pruning, 2023

---

> ### Author Rebuttal · Authors · 2026-03-26
>
> We sincerely thank the reviewer for the constructive feedback. We address your questions and provide clarifications below:
>
> > Q1. Model Naming Consistency
>
> A1:We have conducted a thorough proofread and standardized all instances of "QWen" to "Qwen" throughout the revision.
>
> > Q2.1. Representation Drift Compensation vs. Optimal Brain Compression
>
> A2.1: We appreciate the suggestion. We will add a "Preliminary" section to summarize the problem definition.
>
> OBC method is designed for **unstructured pruning and quantization**. Its error analysis models **how modifying one value affects another within the local weight matrix**.
>
> In contrast, our representation drift theory focuses on the **global network dynamics in low-rank decomposition**. Our theory identifies how these the truncation errors propagate, interact with non-linearities, and exponentially amplify across deep Transformer blocks.
>
> > Q2.2. Respective Distributions of Local and Propagated Error
>
> A2.2: The local error, defined as $WX - W_rX$, is entirely governed by the mean $\mu$ and covariance $\Sigma$ of input $X$ since both $W$ and $W_r$ are static. When propagating through the deep network, these localized errors are non-linearly amplified by the layer-wise Jacobians, accumulating into a substantial propagated error. Empirically, as visualized in Figure 2, this propagated error alters the representation distribution, manifesting as a drastic reduction in the magnitude of deep hidden states.
>
> > Q2.3: Can the propagated error be ignored?
>
> A2.3: Absolutely not. Ignoring the propagated error is the precise reason why vanilla layer-wise decomposition techniques suffer catastrophic performance drops.
>
> > Q2.4 Exactly which type of error is $c$ compensating for?
>
> The vector $c$ explicitly **targets the local error** at its source to **suppress the propagated error** from materializing. By reducing this initial error injection into the computational graph, we effectively "starve" the propagation chain and align the output with the original model.
>
>
> > Q3. Optimization Landscape of the Compensation Vector
>
> A3: We respectfully clarify that the empirical observation of a "well-behaved loss landscape" is a beneficial property of our method, not its fundamental motivation.
> - Motivation and Theoretical Analysis: The motivation for the compensation vector $c$ stems from our theoretical analysis (Section 3 specifically Proposition 3.1), **learning a bias vector $c$ can explicitly bound the reconstruction error**. Furthermore, by freezing the massive pre-trained weights and optimizing only a single, low-dimensional bias vector, we **drastically reduce the parameter search space from $\mathcal{O}(d^2)$ to $\mathcal{O}(d)$** ($d$ is the hidden size, e.g., 4096). This massive reduction **smooths the loss landscape, yielding a well-behaved optimization space**.
> - Empirical Evidence: To empirically validate this stability, we have provided detailed optimization trajectories in Appendix G (Figure 5). As shown, across different architectures (LLaMA-3-8B and LLaMA-2-7B), **the loss drops steeply within the first few batches and converges to a stable minimum**.
>
> > Q4. Why introduce $c$ instead of directly optimizing bias $b$?
>
> A4: Modern LLMs, such as LLaMA and Qwen, employ bias-free linear layers. Therefore, introducing $c$ necessity to ensure our method is universally applicable. Furthermore, defining $c$ isolates this targeted mathematical correction from the original, pre-trained function of the bias parameter, making the fundamental motivation of "drift compensation" much clearer. we will explicitly clarify the necessity in the revision.
>
> > Q5. Why not other compensation options (e.g., compensating A and B)?
>
> A5: Updating $A$ and $B$ involves modifying millions of parameters, which is computationally heavier and prone to overfitting the small calibration set. A bias vector $c$ provides a lightweight, efficient global shift that mitigates the drift without distorting the learned linear projections.

---

> > ### Author Rebuttal · Reviewer_pQu6 · 2026-04-01
> >
> > Thank you for the detailed rebuttal. I appreciate the authors’ efforts, but I still have some concerns.
> >
> > 1.  In A4, the authors state that modern LLMs such as LLaMA and Qwen use bias-free linear layers. However, to my knowledge, the Qwen2.5 family (one of the most widely used recent Qwen variants) includes bias terms in at least some attention projections (e.g., q/k/v in the its huggingFace implementation). More importantly, if some target architectures are indeed bias-free and the method therefore needs to introduce a new additive parameter that is later fused into the linear operator, does the paper’s “zero inference overhead” wording need to be stated more carefully?
> >
> > 2. In A2.2, my original question was about the distributions of the local and propagated errors, not just their moments. Claiming that the local error is "entirely governed by $\mu$ and $\Sigma$ of input X" only describes the first two moments of RX, not its distribution. Are you assuming Gaussianity? If yes, please state and justify this. If no, then $\mu$  and $\Sigma$  alone cannot characterize the distribution, making the optimality claim for $c^* = R\mu$ conditional on unstated assumptions.

---

> > > ### Author Response · Authors · 2026-04-02
> > >
> > > We sincerely thank you for your rigorous follow-up.
> > >
> > > 1. Clarification on Bias-Free Architectures and "Zero Inference Overhead"
> > >
> > > We appreciate your architectural insights. For models with native biases, the overhead is strictly zero. For bias-free models, we will revise our phrasing from "Zero Inference Overhead" to "Negligible or Near-Zero Inference Overhead". We justify this "negligible" claim from two practical perspectives:
> > >
> > > - Zero Latency Overhead: In current hardware implementations, matrix multiplications are executed via GEMM kernels, adding zero inference latency. Introducing a 1D bias vector is natively supported.
> > > - Negligible Storage Overhead: As shown in Appendix Table 10, even for strictly bias-free models where $c$ must be stored as a new parameter, the memory consumed is imperceptible. For instance, in a 13B model (more than 24 GB in Bfloat16), the additional parameters amount to only 6.5 MB—less than 0.03% of the total model size.
> > >
> > > 2. Clarification on Error Distributions and the Optimality Claim
> > >
> > > No, we do not assume Gaussianity, nor do we assume any specific parametric distribution. Our optimality claim for $c^* = R\mu$ rests on a fundamental property of Mean Squared Error ($L_2$ loss) minimization, which is mathematically **distribution-agnostic**.
> > >
> > > For any random variable with finite moments (which naturally holds for bounded LLM activations), the constant $c$ that minimizes the expected squared error $\mathbb{E}[\|RX - c\|_2^2]$ is universally its expected value: $c^* = \mathbb{E}[RX] = R\mu$. When we stated that the error is "governed by $\mu$ and $\Sigma$," we meant that under an $L_2$ objective, the exact shape of the error's distribution does not change the optimal solution.

---

### Official Review · Reviewer_cjjd · 2026-03-03

**Soundness:** 2
**Presentation:** 3
**Significance:** 2
**Originality:** 2
**Overall Recommendation:** 3
**Confidence:** 4

**Summary:**

This paper focuses on the issue of "Representation Drift" in the low-rank decomposition of Large Language Models (LLMs), where approximation errors accumulate and amplify through deep architectures, leading to a significant degradation in model performance.
To address this, the authors propose a compensation mechanism named "Decomper," which introduces learnable bias vectors into the decomposed linear layers to align with the original output distribution, effectively mitigating performance decay and enhancing model recovery capabilities without any additional inference overhead.

**Compliance With Llm Reviewing Policy:**

Affirmed.

**Final Justification:**

After reading the comments from the other reviewers, I believe this paper still has many issues that require revision and likely cannot be adequately addressed within this submission cycle. In addition, it seems that the authors tend to respond to my concerns indirectly rather than addressing them head-on.

For example, I did not see a response to my comment: *“In my view, fine-tuning the weights and biases of linear layers can both serve a compensatory purpose and potentially achieve the effects you are targeting.”* The authors could have simply trained the biases to examine the effect and respond to this point. Similarly, regarding my Question 4, the authors could have easily checked the compensatory effects of their proposed method on approaches such as SVD-LLM and Dobi-SVD.

Overall, I will maintain my original score.

**Key Questions For Authors:**

Although I have raised several weaknesses above, I do acknowledge that the authors conducted extensive experiments. This suggests that they have a deep understanding of the task. My comments were formed within the limited time of a few hours of reviewing, so some oversights on my part are possible. If the authors can adequately address my concerns or provide a thorough discussion, I would be willing to increase my score.

**Limitations:**

yes

**Strengths And Weaknesses:**

**Strength:**

1. The problem discussed by the authors is important, as research on lightweight techniques for large language models is of significant practical value for real-world deployment.

2. The method proposed by the authors, performing distribution compensation on compressed layers, is intuitive and easy for readers to understand.

**Weakness:**

1. The use of the term "Zero-Cost" in the title and throughout the paper is potentially misleading. While the authors argue that the compensation vector can be fused into the original bias term to achieve zero *inference* overhead, the method itself is not "zero-cost" in a broader sense. It requires a calibration dataset, gradient-based optimization, and additional computational resources during the pre-processing stage. To avoid over-hyping the contribution, the authors should use more precise language, such as "Zero-Inference-Overhead," to distinguish it from methods that require no data or no training.

2. The methodological novelty appears to be incremental. The core idea—adding a learnable compensation vector to each linear layer—can be viewed as a simplified or "ultra-lightweight" version of LoRA (Low-Rank Adaptation). Specifically, while LoRA optimizes low-rank matrices ($U$ and $V$), this method essentially optimizes a Rank-1 bias adjustment. The paper lacks a rigorous discussion or empirical comparison to justify why this specific form of compensation is superior to, or fundamentally different from, applying LoRA with an extremely low rank on the decomposed layers.

3. The experimental validation in Table 2 is insufficient. The authors failed to include **FLAP**, a much simpler static compensation method, in the results for LLaMA-3-8B and Qwen-8B. Considering that FLAP's performance (Perplexity) on LLaMA-2-7B is nearly identical to (and in some cases, only marginally lower than) the proposed method, it is crucial to demonstrate whether the complex optimization process of "Decomper" actually yields significant gains over simpler heuristic-based bias compensation across newer architectures.

4. As a compensation strategy rather than a standalone decomposition algorithm, the experimental design should focus more on its generalizability and cumulative gains:

* **Plug-and-play validation:** The authors should evaluate the performance gains when applying "Decomper" to various existing methods (e.g., Method A + Decomper vs. Method B + Decomper) to show its robustness as a general enhancement tool.

* **Compatibility with Post-Compression Tuning:** It is unclear whether the "optimized compensation" provides a better initialization for subsequent fine-tuning. A critical experiment would be comparing the performance of a model with Decomper followed by post-compression tuning versus a standard decomposed model directly undergoing the same amount of post-compression tuning. Existing works including SVD-LLM have mentioned post-compression tuning.

---

> ### Author Rebuttal · Authors · 2026-03-27
>
> We sincerely thank the reviewer for the thoughtful feedback and for recognizing the practical value of our work.
>
> > Q1. "Zero-Cost "  -> "Zero-Inference-Overhead"
>
> A1: We have replaced "Zero-Cost" with "Zero-Inference-Overhead" throughout the title and main text in the revised manuscript.
>
> > Q2. The method resembles a simplified or "ultra-lightweight" version of LoRA. Why is Decomper different from, and superior to, applying an extremely low-rank (Rank-1) LoRA?
>
> A2:
> 1. We respectfully clarify that Decomper is fundamentally different LoRA.
>
> For LoRA:
> - Even an ultra-low Rank-1 LoRA requires optimizing two matrices ($A \in \mathbb{R}^{m \times 1}, B \in \mathbb{R}^{1 \times n}$) and modifies the weight matrix itself.
> - LoRA is designed for end-to-end task adaptation using target labels.
>
> For Decomper:
> - It uses a single vector to correct representation drift caused by low-rank truncation.
> - Decomper is explicitly designed for task-agnostic reconstruction.
>
> 2. Empirical Comparison (Decomper vs. Rank-1 LoRA)
>
> We apply an extremely low-rank (Rank-1) LoRA to the decomposed model. It fails to recover performance (PPL remains stagnant at 9.33, and average accuracy slightly drops), whereas Decomper successfully improve both perplexity and downstream accuracy.
>
> | Method (LLaMA-2-7B, 30% Ratio) | PPL ($\downarrow$) | Avg. Acc. ($\uparrow$) | BoolQ | PIQA | WinoG. | HellaS. | ARC-e | ARC-c | OBQA |
> | :--- | :---: | :---: | :---: | :---: | :---: | :---: | :---: | :---: | :---: |
> | w/o LoRA or Compen. | 9.33 | 0.537 | 0.6229 | 0.6828 | 0.6298 | 0.5439 | 0.5774 | 0.3183 | 0.3820 |
> | Rank-1 LoRA | 9.33 | 0.532 | 0.6232 | 0.6806 | 0.6267 | 0.5409 | 0.5762 | 0.3131 | 0.3600 |
> | Our Compen. | **8.07** | **0.552** | **0.6235** | **0.7029** | **0.6298** | **0.5847** | **0.6044** | **0.3319** | **0.3840** |
>
>
> > Q3. Missing FLAP baselines on newer architectures & superiority over heuristic compensation?
>
> A3:
> 1. Superiority at Scale/Compression: FLAP is competitive only on LLaMA-2-7B PPL at 20%. Decomper dominates at larger scales/ratios, achieving 50.00 Acc vs FLAP's 44.77 (LLaMA-2-7B), and 7.99 PPL vs FLAP's 11.35 (LLaMA-2-13B).
>
> 2. FLAP was omitted strictly because its official repo lacks Grouped Query Attention (GQA) support. We extended it for LLaMA-3-8B. Decomper remains strictly superior:
> | Method (LLaMA-3-8B) | Ratio | PPL ($\downarrow$) | Avg. Acc. ($\uparrow$) | BoolQ | PIQA | WinoG. | HellaS. | ARC-e | ARC-c | OBQA |
> | :--- | :---: | :---: | :---: | :---: | :---: | :---: | :---: | :---: | :---: | :---: |
> | FLAP | 20% | 12.01 | 0.5275 | 0.6374 | 0.6796 | 0.6385 | 0.5329 | 0.5314 | 0.3145 | 0.3580 |
> | **Ours** | 20% | **11.17** | **0.5717** | **0.6269** | **0.6997** | **0.6622** | **0.6060** | **0.6536** | **0.3797** | **0.3740** |
> | FLAP | 30% | 18.65 | 0.4732 | 0.6106 | 0.6126 | 0.5748 | 0.4388 | 0.4784 | 0.2955 | 0.3020 |
> | **Ours** | 30% | **17.26** | **0.5065** | **0.6349** | **0.6507** | **0.6133** | **0.4863** | **0.5189** | **0.3131** | **0.3280** |
>
> 3. Furthermore, we highlight that **applying FLAP's simple averaging compensation to low-rank decomposed matrices is theoretically and empirically insufficient**: it ignores the input-dependent variance term and the subsequent non-linear distortions (c.f. the second-to-last paragraph on Page 4). As shown below, the FLAP-style compensation yields negligible gains:
> | Model (30% Ratio) | Metric | w/o Compen. | FLAP-style Compen. | **Ours** |
> | :--- | :--- | :---: | :---: | :---: |
> | LLaMA-2-7B | PPL ($\downarrow$) | 9.33 | 9.05 | **8.07** |
> | | Avg. Acc. ($\uparrow$) | 0.537 | 0.541 | **0.552** |
> | LLaMA-3-8B | PPL ($\downarrow$) | 29.22 | 27.20 | **16.17** |
> | | Avg. Acc. ($\uparrow$) | 0.473 | 0.479 | **0.507** |
>
>
> > Q4-1 Plug-and-play Validation
>
> A4-1: We have already conducted the plug-and-play evaluations in Appendix D (Table 7). Adding Decomper to AFM (a PCA-based, feature-space decomposition method) on LLaMA-2-7B (40%) **recovers PPL from 18.82 to 10.44** and **boosts accuracy from 42.05 to 50.16**. We will add a stronger pointer in the main text.
>
> > Q4-2 Compatibility with Post-Compression Tuning
>
> A4-2: Yes, our "optimized compensation" provides a better initialization for subsequent fine-tuning. Under the same data and fine-tuning hyperparameters, "optimized compensation" yields a better final performance than fine-tuning a standard decomposed model.
> | Method (LLaMA-2-7B, 30% Ratio) | PPL ($\downarrow$) | Avg. Acc. ($\uparrow$) | BoolQ | PIQA | WinoG. | HellaS. | ARC-e | ARC-c | OBQA |
> | :--- | :---: | :---: | :---: | :---: | :---: | :---: | :---: | :---: | :---: |
> | w/o Compen. (standard decomposition) | 9.33 | 0.537 | 0.6229 | 0.6828 | 0.6298 | 0.5439 | 0.5774 | 0.3183 | 0.382 |
> | w/o Compen. + LoRA Tuning | 8.75 | 0.545 | 0.6226 | 0.7040 | 0.6275 | 0.5736 | 0.5854 | 0.3362 | 0.368 |
> | w/ Compen. + LoRA Tuning | **8.01** | **0.560** | **0.6242** | **0.7144** | **0.6338** | **0.6026** | **0.6082** | **0.3490** | 0.380 |

---

> > ### Author Rebuttal · Reviewer_cjjd · 2026-04-01
> >
> > Thank you for the authors’ response. However, at this stage I do not plan to adjust my score because I still have some concerns. First, Rank-1 LoRA seems to have no effect in your results (Answer 2), while in Answer 4-2 LoRA shows a clear improvement. How should this be understood? Is it because the rank is too low to be effective? I am concerned that there might be some inconsistency or issue here that could affect the rigor of the paper.
> >
> > Second, I am not fully convinced by the discussion in “We respectfully clarify that Decomper is fundamentally different from LoRA.” LoRA is essentially just a fine-tuning technique rather than only for end-to-end task adaptation. In my view, fine-tuning the weights and biases of linear layers can both serve a compensatory purpose and potentially achieve the effects you are targeting. The main contribution of this paper seems to be the empirical finding that modifying the bias terms of linear layers is more efficient than modifying the weights.

---

> > > ### Author Response · Authors · 2026-04-02
> > >
> > > We sincerely thank you for engaging in this discussion. We appreciate the opportunity to clarify these experimental settings and discuss the theoretical depth of our contribution.
> > >
> > > 1. Clarifying the Apparent "Inconsistency": Rank-1 LoRA (A2) vs. LoRA Tuning (A4-2)
> > >
> > > There is no inconsistency in the results; rather, the two answers reflect **two completely different experimental environments and goals**. We sincerely apologize for not explicitly detailing the rank differences in our previous response:
> > >
> > > - **In Answer 2:**
> > >
> > > This experiment specifically addressed your question about an **"ultra-lightweight version of LoRA (Rank-1 LoRA)."** We compared our 1D compensation vector against a Rank-1 LoRA under the same setting.
> > >
> > > **The Key Finding:** A Rank-1 LoRA actually uses more parameters ($A \in \mathbb{R}^{m \times 1}, B \in \mathbb{R}^{1 \times n}$) than our Decomper (e.g., 4096). Despite having a larger parameter, Rank-1 LoRA completely failed to recover performance, whereas Decomper effectively restored it.
> > >
> > > - **In Answer 4-2:**
> > >
> > > This experiment addressed your separate question regarding **"Compatibility with Post-Compression Tuning."** Following your advice and the standard practice in existing works like SVD-LLM, we applied a **Rank-8 LoRA** to both the standard decomposed model and our Decomper-enhanced model.
> > >
> > > **The Key Finding:** Because it is Rank-8 and undergoes standard fine-tuning, it naturally shows clear improvements overall. However, the crucial takeaway is that **Decomper + Rank-8 LoRA** still outperforms **Standard Decomposition + Rank-8 LoRA**, proving Decomper provides a strictly better initialization.
> > >
> > > 2. The Core Contribution: Beyond "Biases vs. Weights"
> > >
> > > We respectfully argue that our contribution goes beyond the empirical finding that "biases are cheaper to update than weights." Our specific contribution is as follows:
> > > - Identify: To our knowledge, we are the first to specifically identify and formalize representation drift as the primary cause of degradation in low-rank decomposed LLMs.
> > > - Verify: We not only mathematically formalize how local truncation errors are non-linearly amplified across deep blocks, but we also visually demonstrate how this drift manifests as a severe magnitude decay in deep hidden states (e.g., Figure 2).
> > > - Solve: Grounded in this theory, we propose Decomper, a zero-inference-overhead remedy that explicitly absorbs the decomposition-induced drift.
> > >
> > > This complete trajectory—from targeted SVD error diagnosis and visual verification to a practical, zero-cost remedy—strictly sets our contribution apart from existing literature.

---

### Official Review · Reviewer_qXbx · 2026-03-11

**Soundness:** 3
**Presentation:** 3
**Significance:** 3
**Originality:** 3
**Overall Recommendation:** 4
**Confidence:** 3

**Summary:**

This paper recognizes a limitation of SVD-based model compression approaches, which the authors call representation drift: the errors due to low-rank decompositions amplify and propagate throughout the layers of a network. The authors propose a remedy to this called Decomper, which adds bias terms in the linear layers to account for the errors in individual Transformer blocks. These bias terms are learned via calibration data post-training, and then fused into the pre-trained bias terms after being learned. The authors include experiments on a variety of models and datasets comparing their approach to other SVD-based model compression methods.

**Compliance With Llm Reviewing Policy:**

Affirmed.

**Final Justification:**

This work provides a simple yet effective approach to mitigate performance degradation in low-rank compression of LLMs. There were initial concerns regarding robustness to weight rank and extreme compression ratios, but the authors' rebuttal addressed these concerns. Thus, I'd like to maintain my initial positive evaluation.

**Key Questions For Authors:**

1. For a single transformer block, is a single compensation term learned and then fused into every bias term, or is a different compensation term learned then fused for every linear layer within a transformer block? (Weakness 1)

2. How robust is the learned compensation to the chosen weight rank? (Weakness 2)

**Limitations:**

One possible limitation is that the authors consider representation drift from the output of whole transformer blocks, and learn a compensation vector for each block. It’s possible a more fine-grained approach would yield better gains; e.g., investigating representation drift in attention and MLP components individually, and applying more fine-grained compensations to those components.

**Strengths And Weaknesses:**

Strengths:
1. The paper is well-written and presents most of its ideas clearly.

2. Extensive experiments across a range of model sizes and datasets show that Decomper effectively addresses representation drift, and is an effective approach for low-rank compression.

3. Ablation studies and empirical analyses show its robustness and compatibility with other approaches, e.g., quantization.

Weaknesses:
1. From the paper, it’s not clear if, for a single transformer block, a single compensation term is learned and then fused into every bias term, or if a different compensation term is learned then fused for every linear layer within a transformer block.

2. Experiments include compression ratios up to 40%, while previous SVD-based compression approaches, e.g., [1], include results for compression ratios up to 80%.

3. There are no experiments on the robustness of the learned $\hat{c}$ vs. the chosen weight matrix rank, which could be important. The authors claim that the loss landscape in Eq. 6 is typically well-behaved, but it would be interesting to see empirically how that changes with different ranks / compression ratios.

[1] https://arxiv.org/abs/2403.07378

---

> ### Author Rebuttal · Authors · 2026-03-26
>
> We sincerely thank the reviewer for their time and positive evaluation of our work. We address your specific questions and provide clarifications below:
>
> > Q1. Granularity of the Compensation Term (Addressing Weakness 1, Question1, and Limitations)
>
> A1: We learn a **distinct** compensation vector for **every individual linear layer** within both the Attention and MLP components (i.e., separate $c$ vectors for $W_q, W_k, W_v, W_o$, as well as $W_{up}, W_{gate}, W_{down}$).
>
> By optimizing the distinct, layer-specific vectors jointly using the block-level output divergence (Equation 6), the vectors cooperatively compensate for the complex, non-linear distortions.
>
> We will update Section 3.2 and Algorithm 1 to make this clear.
>
> > Q2: Experiments on Extreme Compression Ratios (e.g., up to 80%)
>
> A2: We appreciate the reference to the extreme compression ratios explored in SVD-LLM [1]. From the perspective of practical deployment utility, parameter reduction methods suffer catastrophic performance collapse at extremely high compression ratios. For instance, as shown in Table 1 of the SVD-LLM paper, the model's output degrades to **near random guessing at ratios exceeding 40%**. Therefore, we focus our analysis on the 20%–40% regime.
>
> > Q3. Robustness to Weight Rank (Addressing Weakness 3 & Question 2)
>
> We respectfully clarify that in the context of our low-rank decomposition framework, the **chosen weight matrix rank is equivalent to the compression ratio**. A higher compression ratio directly corresponds to a lower chosen rank for the weight matrices.
> As demonstrated in our Ablation Study (**Table 3**), we isolated the effect of the compensation vector $c$ at different ranks (30% and 40% compression ratios):
> - At a higher rank (30% compression), $c$ provides solid recovery (e.g., OPT-13B PPL improves by 45%).
> - At a lower rank (40% compression), where the initial truncation error is far more severe, $c$ prevents catastrophic degradation and yields even more dramatic relative improvements (e.g., OPT-13B PPL improves by 77%, and accuracy increases by 24% relatively).
>
> This confirms that our compensation mechanism consistently and robustly mitigates representation drift regardless of the chosen weight rank.

---

> > ### Author Rebuttal · Reviewer_qXbx · 2026-04-02
> >
> > Thank you for the response - most of my concerns have been addressed. However, I still have concerns regarding extreme compression ratios.
> >
> > The claim that "compensation mechanism consistently and robustly mitigates representation drift regardless of the chosen weight rank" makes it seem like high compression ratios would exactly be a setting where Decomper would exhibit the most benefits. If Decomper can indeed mitigate representation drift robustly w.r.t. the rank, then it should mitigate the catastrophic performance degradations that occur at high ratios compared to other SVD approaches. Because of this, I think focusing on 20 - 40% compression ratios is a bit restrictive, even if other SVD based approaches degrade significantly beyond this regime. Could the authors conduct an experiment showing Decomper's performance on a model and task of their choice at, e.g., 60% compression ratio?

---

> > > ### Author Response · Authors · 2026-04-03
> > >
> > > We sincerely thank you for your continued engagement. Following your advice, we conducted experiments at an extreme 60% compression ratio on both LLaMA-2-7B and LLaMA-3-8B.
> > >
> > > Decomper exhibits its remarkable relative benefits at this high compression ratio. For instance, uncompensated SVD-LLM  completely shatters LLaMA-2-7B PPL by more than 100. Decomper mitigates this massive drift, slashing the perplexity down to 29.18, while providing consistent relative accuracy improvements across downstream tasks.
> > >
> > > | Method | Ratio | PPL ($\downarrow$) | Avg. Acc. ($\uparrow$) | BoolQ | PIQA | WinoG. | HellaS. | ARC-e | ARC-c | OBQA |
> > > | :--- | :---: | :---: | :---: | :---: | :---: | :---: | :---: | :---: | :---: | :---: |
> > > | **LLaMA-2-7B** | | | | | | | | | | |
> > > | w/o Compen. | 60% | 103.67 | 0.353 | 0.3783 | 0.5267 | 0.5138 | 0.2830 | 0.2875 | 0.2227 | 0.2560 |
> > > | **w/ Compen.** | 60% | **29.18** ($\downarrow$ 71.9%) | **0.382** ($\uparrow$ 8.2%) | **0.3804** | **0.5495** | **0.5343** | **0.3217** | **0.3519** | **0.2577** | **0.2780** |
> > > | **LLaMA-3-8B** | | | | | | | | | | |
> > > | w/o Compen. | 60% | 3223.69| 0.341 | 0.3780 | 0.5065 | 0.4909 | 0.2659 | 0.2647 | 0.2228 | 0.2640 |
> > > | **w/ Compen.** | 60% | **157.74** ($\downarrow$ 95.1%)| **0.360** ($\uparrow$ 5.6%) | **0.3783** | **0.5277** | **0.5020** | **0.2698** | **0.3026** | **0.2661** | **0.2700** |

---

### Official Review · Reviewer_xy5z · 2026-03-13

**Soundness:** 3
**Presentation:** 3
**Significance:** 3
**Originality:** 2
**Overall Recommendation:** 4
**Confidence:** 4

**Summary:**

The paper studies the problem of performance degradation after low-rank decomposition of LLMs. They find "representataion drift" as a major cause of this issue since individually small errors accumulate through nonlinear layers, leading to performance degradation. Then, authors propose Decomper, a learned compensation mechanism, to align the output of original and truncated layers.

**Compliance With Llm Reviewing Policy:**

Affirmed.

**Final Justification:**

I believe my initial score (4) is fair considering the method is simple and can be interesting to a subset of ICML audience, with acceptable novelty

**Key Questions For Authors:**

1. In Table 4, it is reported that the performance of Decomper is almost insensible to the number of samples (doen't degrade, doesn't improve). To what extend does is it true? When does degradation start as sample count drops or vice versa?

**Limitations:**

yes

**Strengths And Weaknesses:**

**Strength:**

- Decomper adds no cost during inference as it can be absorbed into the model weights
- Calibration dataset size does not have to be large (512 is claimed to be sufficient) to close the representation gap
- Extensive experiments are provided. The method outperforms baselines across multiple architectures (OPT, LLaMA-2/3, Qwen). For example, at 40% compression on OPT-13B, it reduced perplexity from 68.0 to 15.88.

**Weaknesses:**

- The mathematical notation is harder to follow at times. Especially, randomness isn't defined concretely before talking about expectations.
- Oftentimes Gemma models are harder to manipulate with simple interventions from my experience. Could authors test Decomper on a newer Gemma model at a reasonable size on their preferred set of experiments?
- Although shown to be simple and effective, the novelty is not significant including representation drift. There is a number of theoretical works on information/error propagation over Transformer layers such as [1] (and references therein) and in a more practical works such as [2].

___

**References:**

1. Giorlandino, A., & Goldt, S. (2026). Two failure modes of deep transformers and how to avoid them: A unified theory of signal propagation at initialisation. In Proceedings of the Fourteenth International Conference on Learning Representations (ICLR 2026). https://openreview.net/forum?id=utSqpxQHXq
2. Kuzmin, A., Nagel, M., Van Baalen, M., Behboodi, A., & Blankevoort, T. (2023). Pruning vs quantization: Which is better? In Proceedings of the Thirty-Seventh Conference on Neural Information Processing Systems (NeurIPS 2023). https://openreview.net/forum?id=0OU1ZXXxs5

---

> ### Author Rebuttal · Authors · 2026-03-26
>
> We sincerely thank the reviewer for the positive evaluation. We address your insightful questions and constructive feedback below:
>
> > Q1. Mathematical Notation and Definition of Randomness
>
> A1: We will explicitly define the randomness in the revised manuscript: The "randomness" stems from the input activations $X$. We treat $X$ as a random variable drawn from the empirical distribution $\mathcal{D}$ of the calibration dataset. Consequently, the expectations $\mathbb{E}[\cdot]$ and the statistical moments (mean $\mu$, covariance $\Sigma$) are taken with respect to this empirical distribution $\mathcal{D}$.
>
> > Q2. Generalization to "Hard-to-Manipulate" Architectures: Does your met remain effective on newer Gemma models where simple interventions fail?
>
> A2: We appreciate the suggestion. We evaluated Decomper on the **Gemma-2-9B** model under both 20% and 30% compression ratios.
> As shown below, Decomper successfully yields a substantial recovery, proving its robustness even on complex architectures.
> At a 30% compression ratio, our compensation mechanism **decreases Perplexity from 16.96 down to 11.28** and **boosts Average Accuracy from 0.5501 to 0.6166**.
>
> | Method (Gemma-2-9B) | Ratio | PPL ($\downarrow$) | Avg. Acc. ($\uparrow$) | BoolQ | PIQA | WinoG. | HellaS. | ARC-e | ARC-c | OBQA |
> | :--- | :---: | :---: | :---: | :---: | :---: | :---: | :---: | :---: | :---: | :---: |
> | Dense Model (Original) | 0% | 6.85 | 0.7446 | 0.8428 | 0.8297 | 0.7372 | 0.7994 | 0.8801 | 0.6570 | 0.4660 |
> | SVD-LLM (w/o Compen.) | 20% | 11.46 | 0.6393 | 0.7954 | 0.7410 | 0.6819 | 0.5970 | 0.7597 | 0.4838 | 0.4160 |
> | **Decomper(w/ Compen.)** | 20% | **9.31** | **0.6688** | 0.7713 | **0.7715** | **0.6961** | **0.6876** | **0.8106** | **0.5188** | **0.4260** |
> | SVD-LLM (w/o Compen.) | 30% | 16.96 | 0.5501 | 0.6801 | 0.6768 | 0.6243 | 0.4825 | 0.6545 | 0.3848 | 0.3480 |
> | **Decomper(w/ Compen.)** | 30% | **11.28** | **0.6166** | **0.7131** | **0.7318** | **0.6590** | **0.6058** | **0.7647** | **0.4420** | **0.4000** |
>
> > Q3. Novelty and Connections to Prior Theoretical Work
>
> A3: We thank the reviewer for bringing these excellent papers to our attention. We will certainly add a discussion bridging our representation drift theory with the broader literature:
> - **Distinct Context:** Giorlandino & Goldt (ICLR 2026) provide a unified theory of signal propagation **at initialization** for dense networks, focusing on rank collapse and exploding/vanishing gradients. Kuzmin et al. (NIPS 2023) study error accumulation specifically in the context of **pruning and quantization**.
> - **Our Specific Contribution:**
> 1. **Identify:** To our knowledge, we are the **first** to specifically identify and  formalize **representation drift** as the primary cause of degradation in low-rank decomposed LLMs.
> 2. **Verify:** We not only mathematically formalize how local truncation errors are non-linearly amplified across deep blocks, but we also **visually demonstrate** how this drift manifests as a severe magnitude decay in deep hidden states (e.g., Figure 2).
> 3. **Solve:** Grounded in this theory, we propose Decomper, a **zero-inference-overhead** remedy that explicitly absorbs the decomposition-induced drift.
>
> This complete trajectory—**from targeted SVD error diagnosis and visual verification to a practical, zero-cost remedy**—strictly sets our contribution apart from existing literature.
>
>
> > Q4. Calibration Sample Size Sensitivity
>
> 1. Theoretical Reason for Insensitivity
>
> By freezing the massive pre-trained weights and optimizing only a single, low-dimensional bias vector, we drastically reduce the parameter search space from $\mathcal{O}(d^2)$ to $\mathcal{O}(d)$ ($d$ is the hidden size, e.g., 4096). This massive reduction yields a well-behaved optimization space, enabling rapid stabilization on small datasets.
>
> 2. Empirical Analysis
>
> To empirically demonstrate this, we extended our ablation down to extreme low-sample regimes for Gemma-2-9B, LLaMA-2-7B, and LLaMA-3-8B (at 30% compression).
> - Robust, expected gains are consistently achieved once the sample count reaches ~512.
> - Remarkably, even at an extremely constrained size of just **64 samples**, Decomper achieves massive recovery compared to the uncompensated model (e.g., reducing LLaMA-3-8B PPL from 30.11 to 18.12).
> | Model (30% Ratio) | Metric | w/o Compen. | 64 | 128 | 256 | 512 | 1024 |
> | :--- | :--- | :---: | :---: | :---: | :---: | :---: | :---: |
> | Gemma-2-9B | PPL ($\downarrow$) | 16.96 | 12.31 | 11.88 | 11.61 | 11.71 | 11.28 |
> | | Avg. Acc. ($\uparrow$) | 55.01 | 59.06 | 60.49 | 60.37 | 60.70 | 61.66 |
> | LLaMA-2-7B | PPL ($\downarrow$) | 9.33 | 8.32 | 8.22 | 8.21 | 8.07 | 8.02 |
> | | Avg. Acc. ($\uparrow$) | 53.18 | 54.94 | 55.12 | 55.22 | 55.16 | 55.31 |
> | LLaMA-3-8B | PPL ($\downarrow$) | 30.11 | 18.12 | 17.26 | 16.18 | 16.07 | 14.54 |
> | | Avg. Acc. ($\uparrow$) | 48.66 | 52.01 | 53.07 | 53.29 | 53.53 | 54.07 |

---

> > ### Author Rebuttal · Reviewer_xy5z · 2026-04-01
> >
> > I thank the authors for extensive responses which mostly addressed my concerns. I believe my initial score (4) is fair considering the paper is simple and can be interesting to a subset of ICML audience, but, respectfully, not ground-breaking.
> >
> > Also, just out of curiosity, I believe a comparison between LoRA fine-tuning (with small rank of 1 or 2, and making sure not to increase the rank by using spectral decomposition) and Decomper would be quite valuable on the same calibration size (same optimization objective). It helps readers guide for what size of available data LoRA might be worth extra FT overhead since Decomper seems unable to benefit from more data.

---

> > > ### Author Response · Authors · 2026-04-02
> > >
> > > We sincerely thank you for your continued engagement and for raising this point of curiosity.
> > >
> > > 1. Empirical Comparison (Rank-1 LoRA vs. Decomper)
> > >
> > > You raised an insightful question regarding the comparison between a Rank-1/2 LoRA and Decomper. We conducted this experiment.
> > >
> > > We applied a Rank-1 LoRA to the decomposed model using the same 512 calibration samples. It fails to recover performance (PPL remains stagnant at 9.33, and average accuracy slightly drops). In contrast, Decomper successfully improve both perplexity and downstream accuracy.
> > >
> > > | Method (LLaMA-2-7B, 30% Ratio) | PPL ($\downarrow$) | Avg. Acc. ($\uparrow$) | BoolQ | PIQA | WinoG. | HellaS. | ARC-e | ARC-c | OBQA |
> > > | :--- | :---: | :---: | :---: | :---: | :---: | :---: | :---: | :---: | :---: |
> > > | w/o LoRA or Compen. | 9.33 | 0.537 | 0.6229 | 0.6828 | 0.6298 | 0.5439 | 0.5774 | 0.3183 | 0.3820 |
> > > | Rank-1 LoRA | 9.33 | 0.532 | 0.6232 | 0.6806 | 0.6267 | 0.5409 | 0.5762 | 0.3131 | 0.3600 |
> > > | Our Compen. | **8.07** | **0.552** | **0.6235** | **0.7029** | **0.6298** | **0.5847** | **0.6044** | **0.3319** | **0.3840** |
> > >
> > > 2. We completely agree that providing a practical decision boundary for readers is highly valuable.
> > > - Limited Data/Compute: Decomper is the optimal, zero-inference-overhead choice to structurally correct decomposition drift.
> > > - Abundant Data/Compute: We can use higher-rank LoRA to learn new distributions. In this regime, Decomper serves as the "drift-cleaning" initialization prior to standard LoRA tuning.

---

### Decision · Program_Chairs · 2026-04-30

**Decision:**

Accept (regular)

**Comment:**

This paper formalizes "representation drift" in low-rank LLM decomposition and proposes Decomper, a zero-inference-overhead bias compensation method. Initial concerns regarding misleading terminology, LoRA comparisons, missing baselines, and theoretical depth were substantially resolved through revised terminology, empirical validation at 60% compression, and comparative experiments. Notably, Reviewer pQu6 raised their score from 3 to 4 following the rebuttal, acknowledging the practical value despite lingering theoretical concerns. While Reviewer cjjd maintains reservations on theoretical rigor, the majority of reviewers recognized the empirical effectiveness across diverse architectures, leading to a final recommendation of Weak Accept.